# Screening attendance of breast or cervical cancers and its associated factors among 30–49 year old women in Gedeo zone, South Ethiopia: Cross-sectional study

**Abel Desalegn Demeke**[1,2]*, **Bedilu Deribe**[3,4], **Martha Girma**[3], **Muluken Gizaw**[2,5], **Sefonias Getachew**[2,5], **Susanne Unverzagt**[2,4], **Eva J. Kantelhardt**[2,6,7], **Betty Ferrell**[8], **Eric Sven Kroeber**[2,4☉], **Lesley Taylor**[9☉]

1 Department of Nursing, College of Medicine and Health Sciences, Dilla University, Dilla, Ethiopia, 2 Global and Planetary Health Working Group, Martin-Luther-University Halle-Wittenberg, Halle (Saale), Germany, 3 School of Nursing, College of Medicine and Health Sciences, Hawassa University, Hawassa, Ethiopia, 4 Institute of General Practice and Family Medicine, Center for Health Sciences, Martin Luther University Halle-Wittenberg, Halle, Saxony-Anhalt, Germany, 5 Department of Epidemiology and Biostatistics, School of Public Health, Addis Ababa University, Addis Ababa, Ethiopia, 6 Global and Planetary Health Working Group, Institute for Medical Epidemiology, Biostatistics and Informatics, Center for Health Sciences, Martin-Luther-University Halle-Wittenberg, Halle (Saale), Germany, 7 Department of Gynecology, Medical Faculty, Martin Luther University Halle-Wittenberg, Halle (Saale), Germany, 8 Department of Nursing Education and Research, City of Hope National Medical Center, Comprehensive Cancer Center, Duarte, CA, United States of America, 9 Division of Breast Surgery, Department of Surgery, City of Hope National Medical Center, Comprehensive Cancer Center, Duarte, CA, United States of America

☉ These authors contributed equally to this work.
* adesalegn2513@gmail.com

**Data Availability Statement:** All data files are available online in the Fig share repository at the

## Abstract

### Introduction

Breast and cervical cancers are the most frequent and fatal cancers among women. Thus, early detection is necessary to improve the prognosis of affected women. However, in Ethiopia, the rates of screening remain alarmingly low.

### Objective

To assess the magnitude of screening attendance for breast or cervical cancer, as well as the factors that predict it, among women aged 30–49 years old in Gedeo Zone, South Ethiopia, in 2023.

### Method

A community-based cross-sectional study was conducted using a multi-stage cluster sampling technique. Data were collected using pretested, structured questionnaires by trained interviewers. Univariate and multivariate logistic regression models were employed to identify factors associated with screening attendance.

following link: https://figshare.com/articles/dataset/Dataset_sav/27885822?file=50697831.

**Funding:** The project on which this publication is based was in part funded by the German Federal Ministry of Education and Research 01KA2220B to the NORA Programme. This study was also supported by Else Kroener-Fresenius-Foundation Grant No. 2018_HA31SP. The funders had no role in the study design, data collection and analysis, decision to publish, or preparation of the manuscript.

**Competing interests:** The authors have declared that no competing interests exist.

## Results

A total of 554 women participated in the study. Of them, 132 (23.8%) were screened for breast or cervical cancer. Higher age of 40–44 versus 45–49 years (adjusted odds ratio [AOR] 4.18 [95% CI 1.59, 10.9]), higher education status ([AOR] 5.49 [95% CI 2.01, 13.1]), having family or a friend with history of breast or cervical cancer ([AOR] 5.55 [95% CI 2.47, 12.5]), short anticipated time to seek help ([AOR] 4.66 [1.31, 11.7]), adequate health literacy ([AOR] 6.98 [95% CI 2.82,13.3]) and high self-efficacy ([AOR] 2.32 [95% CI 1.08, 4.96]) were positive factors with higher screening attendance. High response cost ([AOR] 0.19 95% CI [0.08, 0.50]) was a negative factor and associated with lower screening attendance.

## Conclusion and recommendation

The study found that only one in four women attended breast or cervical cancer screening. Screening uptake was better in women with higher education, health literacy, self-efficacy, and older age–similar to factors associated with other health seeking behavior. Interestingly, history of breast or cervical cancer in a friend or relative was also associated with higher uptake. This indicates that in addition to awareness campaigns, personal testimonials of survivors could encourage women to visit screening facilities.

## Introduction

Globally, breast and cervical cancers (BCC) were the two most common cancers diagnosed in women [1]. In 2022, breast cancer was the first and cervical cancer the eighth most diagnosed cancers worldwide, with an estimated number of 2.3 million and 661,021 new cases, while in terms of mortality, these cancers ranked fourth and ninth, with 665,684 and 348,189 deaths in women, respectively [2]. Among African women, in 2020 BCC were the most commonly diagnosed and leading cause of cancer death, with an estimated number of 186,598 new cases and 85,787 deaths for breast cancer and 117,316 new cases and 76,745 deaths for cervical cancer [3]. In Ethiopia in 2019, the leading incident cases and cancer-related mortality in females were cervical cancer (6570 cases, 3870 deaths) and breast cancer (5450 cases, 3700 deaths) [4].

Cancer screening is a secondary prevention strategy that aims for early detection, diagnosis, and treatment. Studies show that early detection of breast cancer through screening activities combined with appropriate treatment may reduce mortality rates due to breast cancer by 25–30% [5], while visual inspection with acetic acid (VIA) reduced the incidence and mortality of cervical cancer by at least 25% [6]. The Ethiopian National Cancer Control Plan recommends clinical breast examinations for women over 18 visiting health institutions and emphasizes promoting breast self-awareness for women at risk. It also recommends population-based cervical cancer screening using VIA for all women aged 30–49 every 5 years [7].

Cancer screening rates are very low in developing countries [8]. Despite efforts to improve access to screening services from governmental and non-governmental organizations in Ethiopia, utilization has not been raised. The national median prevalence of cervical cancer screening was 14.79% [9] and the rate of breast cancer screening ranges between 6.9% [10] and 20.8% [11]. Different studies have documented factors associated with BCC screening utilization worldwide, including the age of the woman, risk perception, financial constraints, marital status, and parity, with a wide variability between studies [12, 13].

In Ethiopia, various studies have indicated low screening attendance, with rates reported at below 30%. These studies have also identified several factors that influence the rate of screening attendance [14–16]. None of these studies have investigated both cancers simultaneously and associated factors using protection motivation theory (PMT). PMT is a theory developed by Rogers in 1975 based on the theory of value expectation to explain: 1) the effects of fear on health attitudes and behaviors, and 2) the effects of fear motivation have a significant impact on behavior choice [17]. By incorporating this theory, our study aims to fill the current knowledge gap by examining factors influencing attendance of breast and cervical cancer screenings in the Gedeo zone, Southern Ethiopia.

This study is the first critical step to improve cancer early detection and aims to fill the gap by assessing the magnitude of screening attendance for breast and cervical cancer and their predictors among women aged 30–49 years old in Gedeo zone, South Ethiopia.

## Methods and materials

The study was conducted in the Gedeo zone, which is located in southern Ethiopia. The zone had eight districts and five towns, with Dilla town as the administrative capital, which is located 359 km from the capital city, Addis Ababa. In 2021, the total population of the zone was 1,247,812, with an estimated crude population density of 774 persons per square kilometer. Gedeo Zone has one teaching hospital, three primary hospitals, 38 health centers, and 146 health posts. Among the health institutions presented in the zones, eight (one teaching hospital, two primary hospitals, and five health centers) provided screening services (source: Gedeo Zonal Health Department).

### Study design, period and population

A community-based cross-sectional study was completed from 29/04/2023 to 28/05/2023. All women of 30–49 years of age living in the Gedeo zone were the source population. For this study, randomly selected women between 30–49 years of age living for at least 6 months in the selected catchment Kebele (neighborhood) with available screening service during the study period who were willing to participate were included in the study. Those who were unable to respond or diagnosed with BCC during the data collection period were excluded.

### Sample size and sampling technique

A single population proportion formula was used to calculate the sample size by considering the following assumptions: The population proportion of cervical cancer screening practice from a previously study would be 22.9% [18] with a 5% margin of error and a 95% confidence level.

Therefore, by using the formula,

$$n = (Z\alpha/2)^2 p(1-p)/d^2$$

Where: n = sample size required; α = level of significance (set at 0.05); Zα/2 = a standardized normal test with α level of significance. It is equivalent to1.96 with the corresponding 95% confidence interval; d = margin of error; p = proportion

Due to multistage sampling of study participants, a design effect of 1.5% and 10% non-response were considered, and the sample size was determined to be 596.

A multi stage cluster sampling technique was used to select the participants. Accordingly, all five-health centers that had screening services were taken as a cluster. Then, 30% of the catchment Kebele (the smallest administrative unit next to a district in Ethiopia) was selected

by a simple random sampling technique from each health center. We assumed that 30% would be a representative proportion. The sample was allocated proportionally, based on the number of households. To obtain the sampling interval (Kth-value), the number of households in each team was divided by the allocated sample for that specific Kebele. Then the starting household was selected randomly, and the next household was selected systematically for every Kth-value. Finally, an eligible participant from each household was interviewed. In households with more than one eligible participant, one was randomly selected using a lottery method. If an eligible participant was not available at the time of data collection, a revisit was made three times.

## Data collection

A structured questionnaire was adopted from different literature [9, 10, 19–23]. The questionnaire was first developed in English, then it was translated to Amharic and Gedeofa and re-translated back in to English by language experts to assure its consistency. Surveys were prepared in the Kobo Collect app. The tool had six parts, including socio-demographic factors (6 items), questions on BCC screening attendance (2 items), BCC health literacy (12 items), reproductive factors (3 items), behavioral factors (18 items), and protection motivation theory constructs (17 items). The validated tools were used to measure health literacy assessed by twelve (12) items using a five point Likert type scale [20]. The 15 items related to symptom knowledge of breast and cervical cancer had three response options (Yes, No, Don't Know). Finally, yes (correct response) was given "1" point and no/don't know (incorrect) was given "0" a maximum of 15 points could be scored [19], and sub constructs of protection motivation theory such as perceived risk by two items, perceived severity by three items, fear arousal by two items, response efficacy by three items, response cost by three items, and self-efficacy by four items [23]. Data was collected by face-to-face interviews. Before data collection, a two day training was given for data collectors by the principal investigator regarding the purpose of the study, contents of questionnaires, method of data collection, and confidentiality and privacy. Questionnaires were pre-tested among 30 participants (5% of total) to check that they were clear, understandable and interpreted as intended. During data collection time, close supervision and monitoring were carried out by the investigator to ensure the quality of the data. Daily evaluation of the data for completeness and any encountered difficulties at the time of data collection was recorded.

## Operational definitions

➢ Women who had been screened either for cervical cancer, breast cancer, or for both were considered to have "screening attendance" of BCC [24].

➢ Women who responded to the health literacy related question and scored the median value or above were considered to have adequate health literacy.

➢ From a total of 15 knowledge-related questions, participants responding less than 50% correct, were categorized as having poor knowledge and greater than 50% as having good knowledge.

➢ Women who reported seeking health care for potential symptoms of BCC for less than one week were considered to have a short anticipated time to seek help [25].

➢ Perceived risk refers to an individual's subjective judgment of the possibility of getting breast or cervical cancer. Perceived severity refers to personal subjective judgment of negative consequences from breast and cervical cancer. Fear arousal refers to an individual's

worry or concern about being affected by BCC. Response efficacy refers to an individual's judgment of the effectiveness of the early detection and treatment of BCC. Response cost referred to an individual's subjective assessment of the costs associated with participating in BCC screening. Self-efficacy refers to individuals' belief in their capabilities and confidence to engage in BCC screening. After a total score was computed for each sub-construct, subjects were dichotomized using the mean score into high (who scored mean or above) and low (who scored below mean) [23].

### Data analysis

IBM SPSS version 25.0 was used for statistical analysis. The data were imported to SPSS from the Kobo Collect app, then recoded, checked, and cleaned. Then, to categorize variables, the distribution was checked. The characteristics of categorical variables were described by a chart and frequency table with frequencies and percentages.

Univariate and multivariate logistic regression analyses were used to identify factors associated with screening attendance towards breast and cervical cancer. In univariate logistic regression analysis, variables fitted at p-value $\leq 0.25$ were entered into multivariate logistic regression after multicollinearity was assessed using a histogram. Finally, multivariate logistic regression analysis was performed, and a p-value of less than 0.05 was considered a strong predictor variable for screening attendance of two female cancers. The model's fitness was checked using Hosmer-Lemeshow goodness-of-fit tests.

### Ethical issues

The study was approved by Hawassa University College of Medicine and Health Sciences Institutional Review Board (IRB) with reference number IRB/293/15. After discussion and explanation about the purpose of the study, written informed consent was obtained from individual study participants just for the actual data collection. They were notified there was no harm in their participation in this study, and they have the right to refuse or terminate at any point of the interview. Confidentiality was assured by making the data collection procedure anonymous.

## Results

### Socio demographic characteristics

Out of 596 selected households in the study, 554 eligible women participated, yielding a response rate of 92.9%. Of the study participants, 41.7% were between 30 and 34 years of age, and 67.5% were from urban areas. The majority of respondents (78.7%) were married. In terms of educational level, more than half of the participants (59.6%) had primary education or below. The majority of participants (41.9%) had an average family monthly income between 1501 and 3000 Ethiopian Birr (**Table 1**).

### Reproductive and behavioral characteristics

A total of 405 (73.1%) of participants responded that they visited health care facilities (pharmacy, clinic, or health center or hospital) with a duration of less than one week, while 3.8% responded that they never visit to seek help for a potential symptom of BCC. The remaining 23.1% responded that they visited a health care facility within a period of one week or greater. Regarding contraceptive usage, 79.8% of women had ever used contraceptives, and 11.4% had no children. Nearly half of participants (49.1%) rated their health status as good; 31% had a family or friend history of BCC (**Table 2**).

**Table 1. Socio-demographic characteristics of women in Gedeo zone, South Ethiopia, 2023 (N = 554).**

| Variable | Total (N = 554) | Screening Attendance (%) | |
|---|---|---|---|
| | | Yes | No |
| Age in years (%) | | | |
| 30–34 | 231(41.7) | 40(30.3%) | 191 (45.3%) |
| 35–39 | 135 (24.4) | 22(16.7%) | 113 (26.8%) |
| 40–44 | 70 (12.6) | 26(19.7%) | 44 (10.4%) |
| 45–49 | 118 (21.3) | 44(33.3%) | 74 (17.5%) |
| Residence (%) | | | |
| Urban | 374 (67.5) | 90 (68.2%) | 284 (67.3%) |
| Rural | 180 (32.5) | 42 (31.8%) | 138 (32.7%) |
| Marital status (%) | | | |
| Divorced/widowed | 62 (11.2) | 24 (18.2%) | 38 (9%) |
| Single | 56 (10.1) | 16 (12.1%) | 40 (9.5%) |
| Married | 436 (78.7) | 92 (69.7%) | 344 (81.5%) |
| Education status (%) | | | |
| Primary education or below | 330 (59.6) | 32 (24.2%) | 298 (70.6%) |
| Secondary education | 104 (18.8) | 58 (48.3%) | 62 (14.7%) |
| College and above | 120 (21.7) | 42 (31.8%) | 62 (14.7%) |
| Employment status (%) | | | |
| Housewife | 191 (34.5) | 18 (13.6%) | 173 (41%) |
| Daily labor | 105 (19) | 42 (31.8) | 63 (14.9%) |
| Merchant | 70 (12.6) | 22 (16.7%) | 48 (11.4%) |
| Private/non-government employee | 101 (18.2) | 7 (5.3%) | 94 (22.3%) |
| Governmental-employee | 87 (15.7) | 43 (32.6%) | 44 (10.4%) |
| Income (%) | | | |
| ≤1500 Birr | 141(25.5%) | 10 (7.6%) | 131(31%) |
| 1501–3000 Birr | 232 (41.9%) | 37(28%) | 195(46.2%) |
| ≥3001 Birr | 181(32.7%) | 85(64.4%) | 96(22.7%) |

## Knowledge related to symptom of BCC

Regarding their knowledge related symptom of BCC from 15 knowledge-related questions, only 30 (5.4%) of participants responded 50% or greater correct answers (had good knowledge), their mean level of knowledge was 1.75 ± 2.98 with minimum and maximum values of 0 and 15, respectively. More than half of the study participants knew that persistent smelly vaginal discharge and only ten percent of participants knew that lumps or thickening in the breast were related to cervical and breast cancer symptoms, respectively (**Table 3**).

A total of 65.2% of participants used mass media (television or radio) as the most commonly used source of health-related information; others named health care providers (30.5%), relatives (6.3%), or friends (12.6%). Only 13.2% of participants named more than one source of information (**Fig 1**).

A total of 45.8% of study participants had a high perceived risk of BCC, and 314 (56.7%) of them had a high perceived severity. A total of 43% of participants had low self-efficacy, and about half of the study participants (50.9%) had a high response cost (**Table 4**).

## Screening attendance and associated factors

Among the total participants, only 132 (23.8%; 95% CI 20% to 27%) had screening attendance for the two-female cancers (screened either for breast or cervical or for both of them). This

**Table 2. Reproductive and behavioral characteristics of women in Gedeo zone, South Ethiopia, 2023 (N = 554).**

| Variables | Total (N = 554) | Screening Attendance (%) | |
| --- | --- | --- | --- |
| | | Yes | No |
| Number of children | | | |
| No | 63 (11.4%) | 27 (49.2%) | 36 (57.1%) |
| 1–2 | 164 (29.6%) | 49 (29.9%) | 115 (70.1%) |
| 3–4 | 189 (34.1%) | 28 (14.8%) | 161 (81.9%) |
| >4 | 138 (24.9%) | 28 (20.3%) | 110 (79.7%) |
| Ever used contraceptive | | | |
| No | 112 (20.2%) | 24 (21.4%) | 88 (78.6%) |
| Yes | 442 (79.8%) | 108(24.4%) | 334 (75.6%) |
| Self-rated health status | | | |
| Poor | 282 (50.9%) | 36 (12.8%) | 246 (87.2%) |
| Good | 272 (49.1%) | 176 (64.7%) | 96 (35.3%) |
| Family/friend history of BC/CC | | | |
| No | 382 (69%) | 85 (22.3%) | 297 (77.7%) |
| Yes | 172 (31%) | 125 (72.7%) | 47 (27.3%) |
| Anticipated time to visit health facility | | | |
| Long | 149 (26.9%) | 7 (4.7%) | 142 (95.3%) |
| Short | 405 (73.1%) | 125 (30.9%) | 280 (69.1%) |

screening attendance was increased with age, in women having family or friend history of BCC, short anticipated time to seek help, women with adequate health literacy, high educational status, high self-efficacy, and low response cost (**Fig 2**).

The odds of screening attendance increased with age: odds were higher among women ages 45–49 years ([AOR] 4.18; [95% CI 1.59 to 10.9]) compared to women ages between 30–34 years. Women having college and above educational status reported higher attendance in comparison with those with primary education and below ([AOR] 5.49; [95% CI 2.01 to 13.1]). Women who had a family history or friend with breast or cervical cancer had higher odds ([AOR] 5.55; [95% CI 2.47 to 12.5]). Similarly, women who had a short anticipated time to seek help had higher odds of screening as compared to those with a long anticipated time ([AOR] 4.66; [95% CI 1.31 to 11.7]). Women who had adequate or sufficient health literacy and high self-efficacy had higher odds for screening attendance ([AOR] 6.98]; [95% CI 2.82 to 13.3] and ([AOR] 2.32 [95% CI 1.08 to 4.96]) in comparison to those with inadequate and low self-efficacy, respectively.

Screening attendance decreased by 81% among women who had a higher response cost in comparison to those with low response cost ([AOR] 0.19; [95% CI 0.08 to 0.50]) (**Table 5**).

## Discussion

The aim of this community-based cross-sectional study was to assess screening attendance of breast or cervical cancer among women aged 30–49 years in Gedeo Zone, southern Ethiopia. The findings provide valuable insights into the current state of cancer screening behavior and identify factors associated with screening attendance.

In the current study, the overall screening rate for at least one of the two most frequent female cancers (breast, cervical, or both) was 23.8%. This low screening attendance highlights the urgent need for information campaigns and educational initiatives to highlight the benefits of early detection. This finding is lower in comparison to studies conducted in China (two studies: 40.6%) [26] and 39.1% [24]). This might be due to the difference in screening program

**Table 3. Frequency of knowledge related to symptoms of breast and cervical cancer among women in Gedeo zone, south Ethiopia 2023 (N = 554).**

| Symptoms of breast cancer | Category | Frequency | Percentage (%) |
|---|---|---|---|
| Change in the size of the nipple | Yes | 112 | 20.2 |
| | No | 138 | 24.9 |
| | I Don't know | 304 | 54.9 |
| Change in the shape of the nipple | Yes | 102 | 18.4 |
| | No | 126 | 22.7 |
| | I Don't know | 326 | 58.8 |
| Nipple rash | Yes | 86 | 15.5 |
| | No | 141 | 25.5 |
| | I Don't know | 327 | 59 |
| Discharge from the nipple | Yes | 136 | 24.5 |
| | No | 139 | 25.1 |
| | I Don't know | 279 | 50.4 |
| Bleeding from the nipple | Yes | 157 | 28.3 |
| | No | 95 | 17.1 |
| | I Don't know | 302 | 54.5 |
| Lump or thickening in the breast | Yes | 54 | 9.7 |
| | No | 119 | 21.5 |
| | I Don't know | 381 | 68.8 |
| **Symptoms of cervical cancers** | **Category** | **Frequency** | **Percentage (%)** |
| Vaginal bleeding between menstrual periods | Yes | 30 | 5.4 |
| | No | 171 | 30.9 |
| | I Don't know | 353 | 63.7 |
| Persistent lower back pain | Yes | 14 | 2.5 |
| | No | 135 | 24.4 |
| | I Don't know | 405 | 73.1 |
| Persistent smelly vaginal discharge | Yes | 352 | 63.5 |
| | No | 86 | 15.5 |
| | I Don't know | 116 | 20.9 |
| Discomfort or pain during sex | Yes | 26 | 4.7 |
| | No | 161 | 29.1 |
| | I Don't know | 367 | 66.2 |
| Menstrual periods that are longer or heavier than usual | Yes | 18 | 3.2 |
| | No | 151 | 27.3 |
| | I Don't know | 385 | 69.5 |
| Persistent lower abdominal or pelvic pain | Yes | 32 | 5.8 |
| | No | 167 | 30.1 |
| | I Don't know | 355 | 64.1 |
| Vaginal bleeding during or after sex | Yes | 51 | 9.2 |
| | No | 118 | 21.3 |
| | I Don't know | 385 | 69.5 |
| Unexplained weight loss | Yes | 11 | 2 |
| | No | 219 | 39.5 |
| | I Don't know | 324 | 58.5 |
| Itching in the vagina | Yes | 26 | 4.7 |
| | No | 194 | 35 |
| | I Don't know | 334 | 60.3 |
| Knowledge | Good | 30 | 5.4 |
| | Poor | 524 | 94.6 |

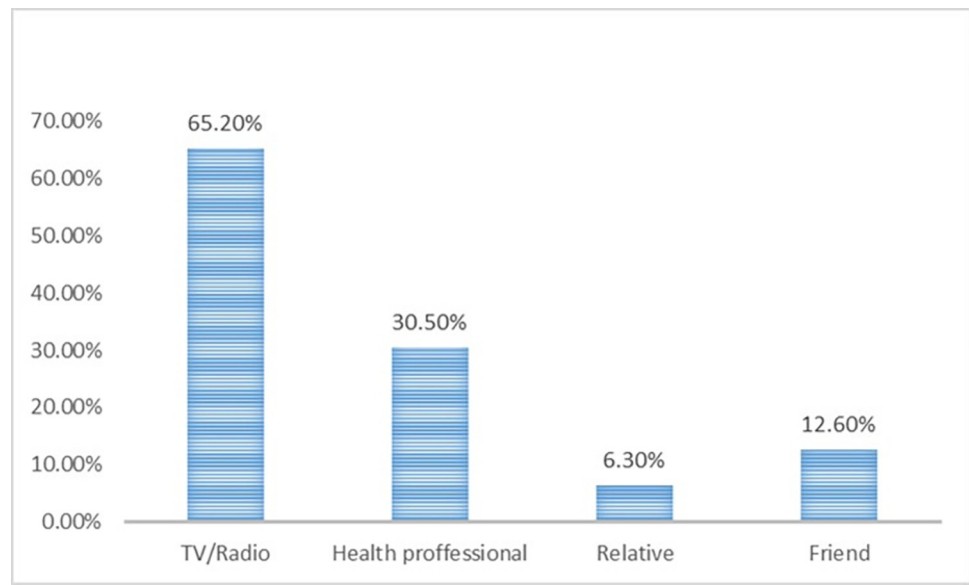

**Fig 1. Health related information source of women in Gedeo zone, south Ethiopia, 2023.**

policy, in which there is an organized screening program for two female cancers in China. Organized screening programs may enable early detection of breast and cervical cancers. Therefore, in Ethiopia, integrating the existing national program of breast cancer screening with cervical cancer screening may be an approach worth exploring for implementation.

Age was found to be associated with BCC screening attendances, which revealed that 45–49 year old women had higher odds of screening attendance than those aged 30–34. This finding is supported by previously conducted studies in Ethiopia [15, 27]. The possible explanation for

**Table 4. Study participant response towards protection motivation theory sub constructs in Gedeo zone, South Ethiopia, 2023 (N = 554).**

| Theory Sub construct | Mean ± SD | Total (%) | Screening attendance | |
|---|---|---|---|---|
| **Perceived risk** | **6.5±2.8** | | **Yes** | **No** |
| Low | | 300 (54.2%) | 97 (32.3%) | 203 (67.7%) |
| High | | 254 (45.8%) | 35 (13.8%) | 219 (86.2%) |
| Perceived severity | 12.7±2.5 | | | |
| Low | | 240 (43.3%) | 59 (24.6%) | 181 (75.4%) |
| High | | 314 (56.7%) | 73 (23.2%) | 241 (76.8%) |
| Fear arousal | 5.9±3.34 | | | |
| Low | | 256 946.2%) | 88 (34.8%) | 168 (65.6%) |
| High | | 298 (53.8%) | 44 (14.6%) | 254 (85.2%) |
| Response efficacy | 10.8±2.6 | | | |
| Low | | 187 (33.8%) | 35 (18.7%) | 152 (81.3%) |
| High | | 367 (66.2%) | 97 (6.4%) | 270 (73.6%) |
| Response cost | 8.9±4.5 | | | |
| Low | | 272 949.1%) | 118 (43.4%) | 154 (56.6%) |
| High | | 282 (50.9%) | 14 (5%) | 268 (95%) |
| Self-efficacy | 14.5±4.5 | | | |
| Low | | 238 (43%) | 41 (17.2%) | 197 (82.8%) |
| High | | 316 (57%) | 91 (28.8%) | 225 (71.2%) |

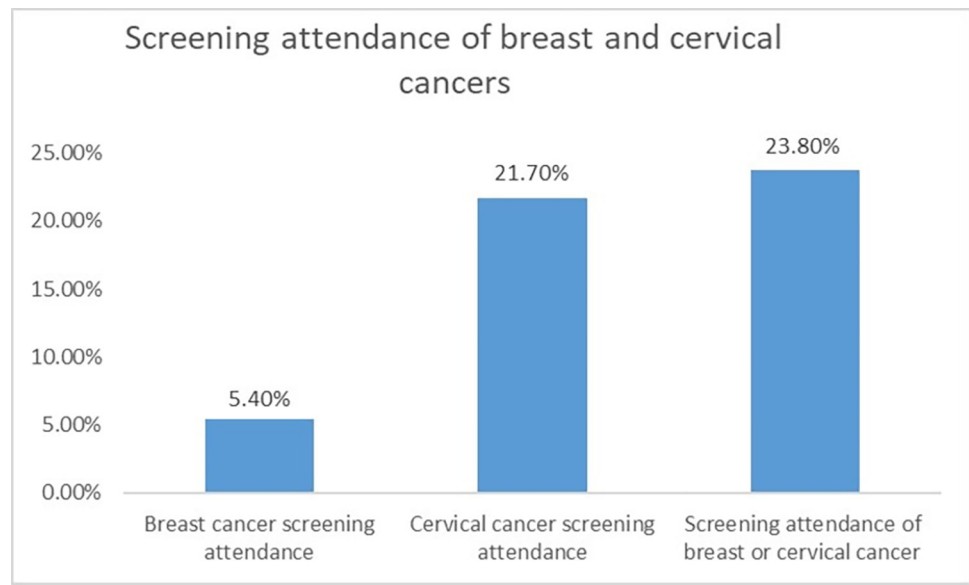

**Fig 2. Breast and cervical cancer screening attendance of women in Gedeo zone, South Ethiopia, 2023 (N = 554).**

this could be that Ethiopian National Cancer Control Plan guidelines support the beginning of screenings at an earlier age [7]. In addition to this, it might be that as age advances, women are believed to have increased risk factors and are more likely to approach health care facilities for different types of reproductive morbidities, which is a practice called opportunistic screening.

This study found that women with higher educational levels had higher screening attendance for BCC. Participants with college and above education levels had five times higher odds of screening attendance compared to those with primary education and below. The finding was consistent with previously conducted studies in other parts of Ethiopia [11, 14] or other countries such as China [26]. Similarly, analysis of WHO studies also showed that higher educational levels significantly improve screening rates in low and middle income countries [28]. Educated women are more likely to have a good understanding of the burden of cancer related disease and its early detection methods.

The study revealed that women with high response costs were less likely to attend screenings of BCC. This indicates that a reduced response cost could facilitate women's engagement in screening and align with a study conducted in Iran [29]. Thus, effective measures should be taken to reduce women's response costs for screening, such as protecting their privacy and giving appropriate advice regarding the two female cancer health practices to reduce their negative feelings.

Having family or friends with breast or cervical cancer increased screening attendance for these female cancers. Previously conducted studies in other parts of Ethiopia and China reported similar associations [11, 23]. A family or friend history of breast or cervical cancer increases awareness and knowledge of the risk of female cancers and accordingly improves their health-seeking behavior. Awareness campaigns can include survivors or relatives giving testimonials to showcase individual stories, encouraging a population that usually considers cancer a death sentence.

Participants who had a short anticipated time to seek help were more likely to have increased odds of screening attendance. In our study, 73.1% of participants had short anticipated time to seek help for possible breast and cervical cancer symptoms; however, only 5.4% of them had good knowledge regarding cancer symptoms. This finding may indicate that to

**Table 5. Factors associated with screening attendance of breast and cervical cancer among women in Gedeo zone, South Ethiopia, 2023 (N = 554).**

| Variables | Screening attendance | | COR | AOR |
|---|---|---|---|---|
| | Yes | No | | |
| **Age** | | | | |
| 30–34 | 40(30.3%) | 191(45.3%) | 1 | 1 |
| 35–39 | 22(16.7%) | 113(26.8%) | 0.93(0.53,1.64) | 0.68(0.29,1.6) |
| 40–44 | 26(19.7%) | 44(10.4%) | **2.82(1.56,5.10)**\* | 1.08(0.41,2.84) |
| 45–49 | 44(33.3%) | 74(17.5%) | **2.84(1.71,4.71)**\* | **4.18(1.59,10.9)** \*\* |
| **Marital status** | | | | |
| Divorced/widowed | 24(18.2%) | 38(9%) | 1 | 1 |
| Single | 16(12.1%) | 40(9.5%) | 0.63(0.29,1.37) | 0.04(0.01,1.39) |
| Married | 92(69.7%) | 344(81.5%) | **0.42(0.24,0.74)**\* | 0.12(0.03,1.42) |
| **Educational status** | | | | |
| Primary education &below | 32(24.2%) | 298(70.6%) | **1** | 1 |
| Secondary education | 58(48.3%) | 62(14.7%) | **6.31(3.69,10.8)**\* | 0.57(0.13,2.46) |
| College and above | 42(31.8%) | 62(14.7%) | **8.71(5.23,14.5)**\* | **5.49(2.01,13.1)**\*\* |
| **Employment status** | | | | |
| Housewife | 18(13.6%) | 173(41%) | 1 | 1 |
| Daily labor | 42(31.8%) | 63(14.9%) | **6.41(3.44,11.9)**\* | 0.23(0.08,1.70) |
| Merchant | 22(16.7%) | 48(11.4%) | **4.41(2.19,8.87)**\* | 1.16(0.39,3.40) |
| NGO/private | 7(5.3%) | 94(22.3%) | 0.72(0.29,1.78) | 2.03(0.56,7.36) |
| Governmental | 43(32.6%) | 44(10.4%) | **9.39(4.94,14.9)**\* | 0.97(0.34,2.83) |
| **Income** | | | | |
| ≤1500 Birr | 10(7.6%) | 131(31%) | 1 | 1 |
| 1501–3000 Birr | 37(28%) | 195(46.2%) | **2.49(1.19,5.17)**\* | 5.52(0.73,12.6) |
| ≥3001 Birr | 85(64.4%) | 96(22.7%) | **11.6(5.72,18.1)**\* | 9.64(0.21,13.2) |
| **Number of children** | | | | |
| None | 27(20.5%) | 36(8.5%) | 1 | 1 |
| 1–2 | 49(37.1%) | 115(27.3%) | 0.57(0.31,1.04) | 0.19(0.03,1.16) |
| 3–4 | 28(21.2%) | 161(38.2%) | **0.23(0.12,0.44)**\* | 0.05(0.01,1.34) |
| >4 | 28(21.2%) | 110(26.1%) | **0.34(0.18,0.65)**\* | 0.22(0.03,1.53) |
| **Family/friend history of BC/CC** | | | | |
| No | 85(64.4%) | 297(70.4%) | 1 | 1 |
| Yes | 47(35.6%) | 125(29.6%) | **1.31(1.17,1.99)**\* | **5.55(2.47,12.5)**\*\* |
| **Number-of information sources** | | | | |
| One | 98(74.2%) | 383(90.8%) | 1 | 1 |
| More than one | 34(25.8%) | 39(9.2%) | **3.41(2.05,5.68)**\* | 0.64(0.22,1.86) |
| **Knowledge** | | | | |
| Poor | 113(85.6%) | 411(97.4%) | 1 | 1 |
| Good | 19(14.4%) | 11(2.6%) | **6.28(2.91,13.6)**\* | 0.96(0.26,3.57) |
| **Health status** | | | | |
| Poor | 36(27.3%) | 246(58.3%) | 1 | 1 |
| Good | 96(72.7%) | 176(41.7%) | **3.73(2.43,5.73)**\* | 1.35(0.61,3.01) |
| **Anticipated time to seek help** | | | | |
| Long | 7(5.3%) | 142(33.6%) | 1 | 1 |
| Short | 125(94.3%) | 280(66.4%) | **9.06(4.12,12.9)**\* | **4.66(1.31,11.7)**\*\* |
| **Health literacy** | | | | |
| Inadequate/problematic | 33(25%) | 305(72.3%) | 1 | 1 |

*(Continued)*

**Table 5.** (Continued)

| Variables | Screening attendance | | COR | AOR |
|---|---|---|---|---|
| | **Yes** | **No** | | |
| Sufficient/Adequate | 99(75%) | 117(27.7%) | **7.82(4.99,12.2)***  | **6.98(2.82,13.3)**** |
| **Perceived risk** | | | | |
| Low | 97(73.5%) | 203(48.1%) | 1 | 1 |
| High | 35(26.5%) | 219(51.9%) | **0.33(0.22,0.52)*** | 0.35(0.16,1.74) |
| **Fear arousal** | | | | |
| Low | 88(66.7%) | 168(39.8%) | 1 | 1 |
| High | 44(33.3%) | 254(60.2%) | **0.33(0.22,0.49)*** | 1.33(0.55,3.21) |
| **Response efficacy** | | | | |
| Low | 35(26.5%) | 152(36%) | 1 | 1 |
| High | 97(73.5%) | 270(64%) | **1.56(1.01,2.41)*** | 0.44(0.18,1.05) |
| **Response cost** | | | | |
| Low | 118(89.4%) | 154(36.5%) | 1 | 1 |
| High | 14(10.6%) | 268(63.5%) | **0.07(0.04,0.12)*** | **0.19(0.08,0.50)**** |
| **Self-efficacy** | | | | |
| Low | 41(31.1%) | 197(46.7%) | 1 | 1 |
| High | 91(68.9%) | 225(53.3%) | **1.94(1.28,2.94)*** | **2.32(1.08,4.96)**** |

COR-crude odd ratio; AOR-adjusted odds ratio

* = P-value ≤ 0.25

** = P-value < 0.05, 1 = reference category

enhance early detection, behavioral interventions will need to be focused on increasing knowledge of symptoms.

Self-efficacy was also found to be associated with screening attendance at BCC. Another study from Iran showed that women with higher self-efficacy were more likely to perform screening regularly [30]. Additionally, the effectiveness of HPV self-sampling and sexual health education in promoting cervical cancer screening among hard-to-reach women emphasizes the importance of self-efficacy in screening uptake, as highlighted by a study conducted in India [31]. This indicates that more attention on self-efficacy in intervention programs is needed to increase women's screening attendance.

Health literacy was associated with screening attendance for two female cancers. Adequate levels of health literacy are associated with increased participation in behaviors that promote health (e.g. screening attendance) while inadequate levels of health literacy are associated with increased participation in health risk behaviors (e.g., screening avoidance). This is consistent with results from previously done studies in India and Australia showing that individuals with lower health literacy may experience greater barriers to adopting protective behaviors due to their limited ability to use health information [32, 33]. Other studies indicate that higher health literacy is associated with favorable health behaviors [34, 35].

This study had a number of limitations and strengths. Our study was based on the self-reported information provided by the participants, which may have led to some information bias. Participants may have provided socially desirable responses. Also, this study only included participants who lived in catchment Kebele's presented under a health center that provides screening services. However, this study was a community-based study with coverage of both urban and rural areas. This study adds evidence to research on non-communicable diseases in a country like Ethiopia, where there is more focus on communicable diseases, and in

exploring a relatively new arena of understanding for the early detection of two female cancers.

## Conclusion

The findings of this study revealed low attendance rates for breast or cervical cancer screenings among women in southern Ethiopia. There is a need for increased efforts to promote and improve screening rates to meet the WHO goals for the global breast and cervical cancer initiatives.

Being older, having higher educational status, family or friend history of breast or cervical cancer, short anticipated time to seek help, adequate health literacy, and high self-efficacy were positive factors, while high response cost was a negative factor associated with screening attendance.

Health care providers should improve health education and motivate age-eligible women using protection motivation theory in the health facility or in the community. Survivors could be included in awareness campaigns for better understanding as well as to show that cancer screening and treatment may lead to overcoming the diseases. Regional, zonal, and district health administrators should enforce public health education and awareness creation by using their most commonly used source of health related information. The Ministry of Health should design health literacy focused screening policy guidelines.

## Acknowledgments

The authors would like to thank Hawassa University College of Medicine and Health Sciences, School of Nursing, Dilla University, Martin Luther–University Halle-Wittenberg from Germany and City of hope from the USA for supporting this research work. Also we would like to extend a special thanks to the Gedeo Zonal Health office for the information they provided to do my work. Furthermore, we would like to thank the study participants; their willingness was invaluable for the success of this work.

## Author Contributions

**Conceptualization:** Abel Desalegn Demeke.

**Formal analysis:** Abel Desalegn Demeke.

**Methodology:** Abel Desalegn Demeke, Bedilu Deribe.

**Supervision:** Eric Sven Kroeber, Lesley Taylor.

**Writing – original draft:** Abel Desalegn Demeke.

**Writing – review & editing:** Martha Girma, Muluken Gizaw, Sefonias Getachew, Susanne Unverzagt, Eva J. Kantelhardt, Betty Ferrell, Eric Sven Kroeber, Lesley Taylor.

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
