## [Decision Letter · Decision Letter 0]

16 Oct 2024

PONE-D-24-16417PLOSE ONE

Screening Attendance of Breast or Cervical Cancers and Its Associated Factors among 30-49 Year Old Women in Gedeo Zone, South Ethiopia: cross-sectional StudyPLOS ONE

Dear Dr. Demeke,

Thank you for submitting your manuscript to PLOS ONE. After careful consideration, we feel that it has merit but does not fully meet PLOS ONE’s publication criteria as it currently stands. Therefore, we invite you to submit a revised version of the manuscript that addresses the points raised during the review process. Please consider all reviewers' comments. Particularly, those regarding typos and English improvements. For example, in tables with missing information, you can describe these frequencies as "No Available-NA" instead of "Don't know". The article includes relevant information to characterize the early-onset population in South Ethiopia, but it is important to represent this data in a language that ensures their spread.

We look forward to receiving your revised manuscript.

Kind regards,

Alexis G. Murillo Carrasco

Academic Editor

PLOS ONE

Journal Requirements:

2. We note that you have indicated that there are restrictions to data sharing for this study. PLOS only allows data to be available upon request if there are legal or ethical restrictions on sharing data publicly. For more information on unacceptable data access restrictions, please see http://journals.plos.org/plosone/s/data-availability#loc-unacceptable-data-access-restrictions. Before we proceed with your manuscript, please address the following prompts: a) If there are ethical or legal restrictions on sharing a de-identified data set, please explain them in detail (e.g., data contain potentially identifying or sensitive patient information, data are owned by a third-party organization, etc.) and who has imposed them (e.g., a Research Ethics Committee or Institutional Review Board, etc.). Please also provide contact information for a data access committee, ethics committee, or other institutional body to which data requests may be sent. b) If there are no restrictions, please upload the minimal anonymized data set necessary to replicate your study findings to a stable, public repository and provide us with the relevant URLs, DOIs, or accession numbers. For a list of recommended repositories, please see https://journals.plos.org/plosone/s/recommended-repositories. You also have the option of uploading the data as Supporting Information files, but we would recommend depositing data directly to a data repository if possible. We will update your Data Availability statement on your behalf to reflect the information you provide.

4. We note you have included a table to which you do not refer in the text of your manuscript. Please ensure that you refer to Table 1-5 in your text; if accepted, production will need this reference to link the reader to the Table.

Reviewers' comments:

Reviewer's Responses to Questions

**Comments to the Author**

1. Is the manuscript technically sound, and do the data support the conclusions?

Reviewer #1: Yes

Reviewer #2: Yes

Reviewer #3: Yes

2. Has the statistical analysis been performed appropriately and rigorously? 

Reviewer #1: N/A

Reviewer #2: N/A

Reviewer #3: Yes

3. Have the authors made all data underlying the findings in their manuscript fully available?

Reviewer #1: No

Reviewer #2: Yes

Reviewer #3: No

4. Is the manuscript presented in an intelligible fashion and written in standard English?

Reviewer #1: No

Reviewer #2: Yes

Reviewer #3: No

5. Review Comments to the Author

Reviewer #1: Dear Authors,

Thank you for your efforts. This is my review:

- Title: Please, remove "PLOSE ONE".

- Abstract-Conclusion: the main factors and relations should be enhaced in the conclusions.

- Methods: the questionnaire is not clear for the reader, you only mention the number of items. To have an idea of the main questions, please add the questionnaire as Supplementary File.

Did you use validated scales? Cite them with original authors.

It is specially interesting to show the questions for protection motivation theory sub constructs.

Please mention the possible bias that derived from a face-to-face interview.

The Operational Definitions should be ordered in bullet points.

- Results: Correct the typo in line174, 186, 196, 207, 230.

Subsection: line211 Screening attendance and associated factors. I recommend to increase the narrative description of the factors which decreased screening attendance: "Screening attendance decreased by 81% among women who had a higher response cost ([AOR] 0.19; [95% CI 0.08 to 0.50]) continue this exposition....

- Discussion: There are much paragraphs which start with "In this study..." "the findings" or similar. Please, vary these introductions to ideas.

Ideas for improving the screening rates. I recommend to introduce a final paragraph with ideas, suggestions or examples of screening programs to try to solve the aforementioned problematic factors (national screening program, health education and knowledge, response costs, self-efficacy in intervention programs,

- References:

REf.18 is incorrectly cited in the list.

Reviewer #2: This study was conducted on breast or cervical cancer screening and its predictors among Ethiopian women, the results of which can be useful for similar communities. The study has a good writing quality, I congratulate the authors for conducting the study, it is hoped that a similar studies will be conducted by screening other related factors.

Some references are missing, they should be corrected.

Reviewer #3: Dear authors thank you for your effort. This is my review for the manuscript " Screening Attendance of Breast or Cervical Cancers and Its Associated Factors among 30-49 Year Old Women in Gedeo Zone, South Ethiopia: cross-sectional Study"

- line 94 in method section: please mention the number of the women of 30-49 years age living in Gedeo zone.

- please include the formula used for calculating sample size.

- the sampling technique isn't clear for me; did you include the attendee of the selected health centers in your study? or the household in catchment Keeble ? or from both?. Also' you mention that "from each health center

30% of catchment Keeble was selected", why you specify 30%?

- Add future clear parctical implication of the study's results as recoomendations at the conclusion section.

- gramatical revision and paraphrasing is required

6. PLOS authors have the option to publish the peer review history of their article (what does this mean?). If published, this will include your full peer review and any attached files.

Reviewer #1: No

Reviewer #2: No

Reviewer #3: No

---

## [Author Response · Author response to Decision Letter 0]

22 Nov 2024

Dear Editorial Team,

Thank you for the opportunity to revise our manuscript titled “Screening Attendance of Breast or Cervical Cancers and Its Associated Factors Among 30-49 Year-Old Women in Gedeo Zone, South Ethiopia: A Cross-Sectional Study.”

We have carefully addressed the comments and suggestions provided by the reviewers. We have uploaded the following files:

Response to Reviewers

Revised Manuscript with Track Changes

Manuscript

Supporting Information (labeled as English Version Questionnaire)

Furthermore, the updated statement of financial disclosure has been included in the cover letter.

We believe these revisions have strengthened our manuscript and effectively addressed the reviewers' concerns. 

Thank you for considering our revised submission. We look forward to your feedback.

Best regards,

Abel Desalegn Demeke

---

## [Decision Letter · Decision Letter 1]

3 Dec 2024

Screening attendance of breast or cervical cancers and its associated factors among 30-49 year old women in Gedeo Zone, South Ethiopia: Cross-sectional study

PONE-D-24-16417R1

Dear Dr. Demeke,

We’re pleased to inform you that your manuscript has been judged scientifically suitable for publication and will be formally accepted for publication once it meets all outstanding technical requirements.

Kind regards,

Alexis G. Murillo Carrasco

Academic Editor

PLOS ONE

Additional Editor Comments (optional):

Reviewers' comments:

Reviewer's Responses to Questions

**Comments to the Author**

1. If the authors have adequately addressed your comments raised in a previous round of review and you feel that this manuscript is now acceptable for publication, you may indicate that here to bypass the “Comments to the Author” section, enter your conflict of interest statement in the “Confidential to Editor” section, and submit your "Accept" recommendation.

Reviewer #1: All comments have been addressed

Reviewer #2: (No Response)

Reviewer #3: All comments have been addressed

2. Is the manuscript technically sound, and do the data support the conclusions?

Reviewer #1: Yes

Reviewer #2: Yes

Reviewer #3: Yes

3. Has the statistical analysis been performed appropriately and rigorously? 

Reviewer #1: Yes

Reviewer #2: Yes

Reviewer #3: Yes

4. Have the authors made all data underlying the findings in their manuscript fully available?

Reviewer #1: (No Response)

Reviewer #2: Yes

Reviewer #3: Yes

5. Is the manuscript presented in an intelligible fashion and written in standard English?

Reviewer #1: (No Response)

Reviewer #2: Yes

Reviewer #3: Yes

6. Review Comments to the Author

Reviewer #1: (No Response)

Reviewer #2: Dear authors

Congratulations on your study. I hope you will continue to research and contribute to the working community.

All comments is corrected.

Reviewer #3: Dear authors, thank you for responding and addressing all the comments and making the required changes

It's a valuable research and I hope it will add a great value to the scientific field

Good luck

7. PLOS authors have the option to publish the peer review history of their article (what does this mean?). If published, this will include your full peer review and any attached files.

Reviewer #1: No

Reviewer #2: No

Reviewer #3: No

---

## [Editor Report · Acceptance letter]

3 Jan 2025

PONE-D-24-16417R1 

PLOS ONE

Dear Dr. Demeke, 

I'm pleased to inform you that your manuscript has been deemed suitable for publication in PLOS ONE. Congratulations! Your manuscript is now being handed over to our production team.

Kind regards, 

on behalf of

Dr. Alexis G. Murillo Carrasco 

Academic Editor

PLOS ONE